# Msn2/4 regulate expression of glycolytic enzymes and control transition from quiescence to growth

Zheng Kuang[1,2,3†], Sudarshan Pinglay[1,2], Hongkai Ji[3]*, Jef D Boeke[1,2]*

[1]Institute for Systems Genetics, NYU Langone Medical Center, New York, United States; [2]Department of Biochemistry and Molecular Pharmacology, NYU Langone Medical Center, New York, United States; [3]Department of Biostatistics, Johns Hopkins University School of Public Health, Baltimore, United States

**Abstract** Nutrient availability and stresses impact a cell's decision to enter a growth state or a quiescent state. Acetyl-CoA stimulates cell growth under nutrient-limiting conditions, but how cells generate acetyl-CoA under starvation stress is less understood. Here, we show that general stress response factors, Msn2 and Msn4, function as master transcriptional regulators of yeast glycolysis via directly binding and activating genes encoding glycolytic enzymes. Yeast cells lacking Msn2 and Msn4 exhibit prevalent repression of glycolytic genes and a significant delay of acetyl-CoA accumulation and reentry into growth from quiescence. Thus Msn2/4 exhibit a dual role in activating carbohydrate metabolism genes and stress response genes. These results suggest a possible mechanism by which starvation-induced stress response factors may prime quiescent cells to reenter growth through glycolysis when nutrients are limited.

DOI: https://doi.org/10.7554/eLife.29938.001

*For correspondence:
hji@jhu.edu (HJ);
jef.boeke@nyumc.org (JDB)

Present address: †Department of Immunology, University of Texas Southwestern Medical Cente, Dallas, United States

Competing interests: The authors declare that no competing interests exist.

## Introduction

Cell growth and proliferation are actively coordinated with extrinsic nutrient availability and intrinsic metabolic states (*Broach, 2012*). When nutrients are limited, cells enter into quiescent states to enhance survival (*Gray et al., 2004*). Repletion of nutrients stimulates quiescent cells back into growth (*Dechant and Peter, 2008*). Gene expression and metabolite profiles are dramatically remodeled during the transitions between quiescence and growth (*Klosinska et al., 2011*). Many transcription factors (TFs), enzymes and metabolites have been shown to regulate the transitions. However, gaps remain in understanding the crosstalk between transcriptional and metabolic activities during the transitions and the mechanisms by which cells make the decision.

Here, we exploited the yeast metabolic cycle (YMC) to study how transcriptionally regulated metabolism impacts cell growth program (*Tu et al., 2005*; *Tu et al., 2007*). In the YMC, cells are synchronized and exhibit respiratory oscillations under continuous, glucose-limited condition (*Figure 1A*). Thousands of transcripts and hundreds of metabolites are cycling as a function of the oxygen consumption oscillations, which divides the YMC into three phases: oxidative (OX), reductive building (RB) and reductive charging (RC). In the OX phase, respiration peaks and growth genes including ribosomal and amino acid biosynthetic genes are activated. Cell division occurs in the RB phase and cell cycle genes and mitochondrion genes are expressed. The RC phase is associated with minimal respiration and genes associated with stress/survival responses, glycolysis and fatty acid oxidation are up-regulated. OX and RB phases can be likened to the growth and proliferation phases based on the burst of respiration and protein synthesis, visualized cell division and the transcriptional and metabolic signatures. RC phase cells exhibit stationary and quiescent characteristics including increased cell density, accumulation of storage carbohydrates glycogen and trehalose and

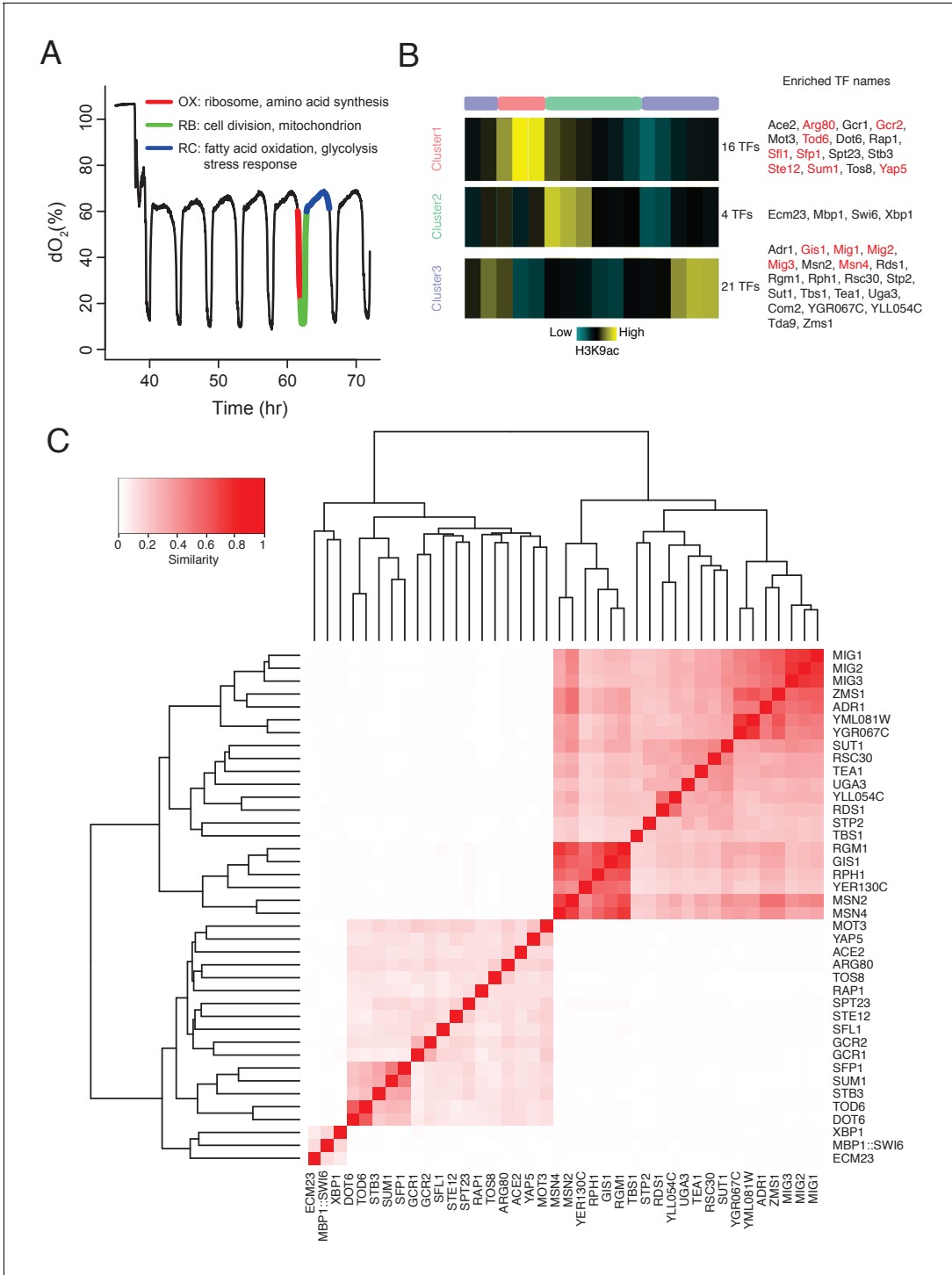

**Figure 1.** DynaMO analysis in YMC identifies key TFs driving YMC. (**A**) Respiratory oscillations (dissolved O2 concentration) of prototrophic yeast strain under continuous nutrient limited condition with a period of 4 ~ 5 hr. Three transcriptional and metabolic different phases are defined: oxidative (OX), reductive/building (RB) and reductive/charging (RC), which are marked by red, green and blue curves. The color scheme is used for labeling the three phases throughout the paper. (**B**) Clustering of predicted motif sites based on temporal H3K9ac signals. TFs with their motif sites enriched in clusters are listed to the right of the heat map. TFs expression which is correlated with corresponding clusters are marked in red. Similar data are presented for completeness in the companion paper by *Kuang et al. (2017) Figure 2A*. (**C**) Similarity of predicted targets of predicted important TFs. Similarity is defined by the ratio of common targets to total targets of two TFs.

DOI: https://doi.org/10.7554/eLife.29938.002

The following source data and figure supplement are available for figure 1:

*Figure 1 continued on next page*

*Figure 1 continued*

**Source data 1.** Similarity scores of predicted targets of predicted important TFs.
DOI: https://doi.org/10.7554/eLife.29938.004
**Source data 2.** GO analysis for predicted targets of each predicted important TFs.
DOI: https://doi.org/10.7554/eLife.29938.005
**Figure supplement 1.** Hierarchical clustering of enriched GO terms based on Fisher Exact test derived FDRs for predicted targets of each predicted important TFs shown in *Figure 1B*.
DOI: https://doi.org/10.7554/eLife.29938.003

expression of stationary specific genes (*Shi et al., 2010*). Therefore, the YMC can be viewed as consecutive alternations of growth, proliferation and quiescent phases. By exploring the oscillations of transcription and metabolites, we may understand the mechanisms through which the cells switch the growth programs. Acetyl-CoA has been shown to drive the transition from RC/quiescence to OX/growth and RB/proliferation phase via the acetylation of histones at growth genes and G1 cyclin CLN3 (*Cai et al., 2011*; *Shi and Tu, 2013*). The role of promoting growth and proliferation by acetyl-CoA has further been shown in various mammalian systems including embryonic stem cells and cancer cells (*Comerford et al., 2014*; *Moussaieff et al., 2015*; *Sutendra et al., 2014*). However, how yeast cells accumulate acetyl-CoA in the quiescent phase to reach the critical level for growth is unclear.

Following this rationale, we developed a computational algorithm, DynaMO, to systematically predict which of the 175 yeast TFs that are linked to specific phases of the YMC (*Kuang et al., 2017*). Briefly, we grouped predicted binding sites for all TFs with similar dynamic binding patterns and for each pair of TF and binding pattern, we examined whether binding sites of the TF are enriched in the cluster/phase (*Figure 1B*). We identified 41 TFs that were specifically linked to each of the three phases in the YMC. We focused on two homologous TFs linked to the RC/quiescent phase, Msn2 and Msn4. They are known to regulate the general stress response in budding yeast, including starvation, thermal, osmotic and oxidative stresses (*Estruch and Carlson, 1993*; *Estruch, 2000*; *Martínez-Pastor et al., 1996*; *Schmitt and McEntee, 1996*). Deleting the two TFs caused a striking delay of transition from RC/quiescence to OX/growth phase in the YMC and slow accumulation of acetyl-CoA. Transcriptomic and cistromic analyses revealed that Msn2/4 control the expression of glycolytic pathway, which may promote the accumulation of acetyl-CoA and re-entry into growth.

## Results

### DynaMO identifies TFs associated with specific phases of the YMC

Histone modification marks transcription regulatory regions and the intensity and dynamic nature of some modifications, such as H3K9ac correlates strongly with transcription activity. Therefore, we can predict TF binding sites by 'foraging' for TF consensus sequences associated with histone modification peaks and then predict the temporal activities at these binding sites in a highly dynamic process like the YMC. We can also identify important TFs associated with specific dynamic programs by enrichment analysis. The functionality has been incorporated in a computational tool, DynaMO (*Kuang et al., 2017*). We used the previously generated 16 time point H3K9ac ChIP-seq data across one round of the YMC (*Kuang et al., 2014*) to predict the binding activities of 175 yeast TFs. Three temporal binding patterns were captured, consistent with the three metabolic phases of the YMC (*Figure 1B*). 41 TFs were identified with their binding sites specifically enriched in one of the three phases, which were thus considered to be candidate regulators of those specific phases. Target genes of each TF were predicted and TFs were clustered by the similarity of targets between pairs of TFs (*Figure 1C*). GO term analysis of predicted targets genes of the 41 TFs reveals similar target specificities among TFs from the same clusters (*Figure 1—figure supplement 1*). In general, the functions of these candidate TFs were consistent with the transcriptional diagram in each phase (*Kuang et al., 2017*), such as ribosome biogenesis TFs Rap1 and Sfp1 enriched in cluster 1 (OX/growth), cell cycle TFs Mbp1 and Swi6 enriched in cluster 2 (RB/proliferation), and stress response TFs Msn2 and Msn4 enriched in cluster 3 (RC/quiescence) (*Figure 1*). Validation experiments

performed by mutating individual TFs also show a significantly higher frequency of disrupted YMC oscillation (as measured by $O_2$ consumption) in the candidate TFs (Arg80, Gcr1, Xbp1, Msn2, Msn4) than in control randomly selected TFs (Skn7, Arg81, Oaf1, Cin5, Ume6). (Various phenotypic defects were observed; the details can be found in *Kuang et al., 2017*) This indicates that we have a list of candidate TFs that help drive the YMC.

## Msn2/4 function in the RC phase to control the transition from the RC/quiescent state to the OX/growth state

From the validation experiments, we observed a very unusual and specific 'lengthened RC phase' phenotype in the *msn2Δ* mutant (*Figure 2A*). The RC phase is longer than that in the WT strain and it gets longer after each cycle. *msn2Δmsn4Δ* showed a more severe defect than *msn2Δ* whereas *msn4Δ* was relatively normal, in support of partial functional redundancy in this instance (*Estruch, 2000*). This may be consistent with previous observations that some genes are completely turned off in the *msn2Δ* single mutant while other genes show reduced expression in the *msn2Δ* single mutant and are completely eliminated in the double mutant (*Estruch, 2000*). The timing of the OX/growth and RB/proliferationΔ phases is normal in the mutants but the RC/quiescence phase is seemingly ever-lengthening, as though the cells await an ever-diminishing signal to proceed. Thus, the prolonged RC phase phenotype observed for *MSN2* and *MSN4* mutants can also be viewed as a delayed transition from RC to OX.

Two lines of evidence suggest that Msn2/4 function in the RC/quiescence phase. First, many stress response genes known to be the targets of Msn2/4 are activated in the RC phase (*Gasch et al., 2000*; *Martínez-Pastor et al., 1996*; *Schmitt and McEntee, 1996*). Second, Msn2/Msn4 motif sites were enriched in the predicted RC phase binding sites (*Figure 1B*, cluster 3). To test this hypothesis, we examined gene expression data for the YMC from the previous study (*Kuang et al., 2014*). *MSN4* mRNA level is increased in the RC phase while *MSN2* mRNA level is relatively constant (*Figure 2B*). This is in line with previous findings that *MSN4* expression is activated by stress but *MSN2* is constitutively expressed (*Gasch et al., 2000*). TFs function by binding to the promoters and enhancers of target genes so we performed time-course ChIP-seq of Msn2 and Msn4 across one YMC. Both TFs showed increased binding to the genome in the RC/quiescence phase regionally (*Figure 2C*) and genome-wide (*Figure 2D*). Msn2 and Msn4 shared a significant proportion of binding sites and target genes (*Figure 2E*, *Figure 2—source data 1* and *2*), supporting their largely redundant functions. Among the targets of Msn2 and Msn4, we observed a substantial proportion of genes expressed in the RC phase (*Figure 2F*), further supporting the hypothesis that Msn2/4 function in RC phase. Additionally, Msn2/4 target genes are enriched in classes of carbohydrate metabolism (*Figure 2F*), indicating that Msn2/4 may function through remodeling the metabolic pathways.

## Msn2/4 regulate expression of the glycolytic pathway and accumulation of acetyl-CoA

Next we investigated how Msn2 and Msn4 control the transition from RC/quiescent state to OX/growth state. We hypothesized that two types of signals may regulate this transition, stimulating signals such as nutrient and inhibitory signals such as stress and damage. Cai *et al*. (*Cai et al., 2011*) showed that adding acetate in the RC phase can directly induce an OX phase and they found that acetyl-CoA is the key metabolite that initiates cell growth and proliferation by promoting histone acetylation at growth genes. Acetyl-CoA is low at the beginning of RC phase and it is accumulated during the RC phase. When it reaches a threshold, it initiates the cell growth program. We asked whether acetate could induce an OX/growth phase in the RC/quiescence phase of a *msn2Δ* mutant. Although the RC phase is much longer in the mutant than that in the WT, acetate still stimulated the OX phase efficiently no matter when in the RC phase acetate was added (*Figure 3A*), consistent with the possibility that acetate or acetyl-CoA represented the limiting factor. Similar results were observed in the *msn2Δmsn4Δ* double mutant. The observations suggested that Msn2 and Msn4 might be involved in nutrient signaling, perhaps based on the accumulating intracellular acetyl-CoA level.

To test this hypothesis, we collected seven time points of WT cells across one YMC and the same seven time points of *msn2Δmsn4Δ* double mutant cells plus four additional time points in the RC

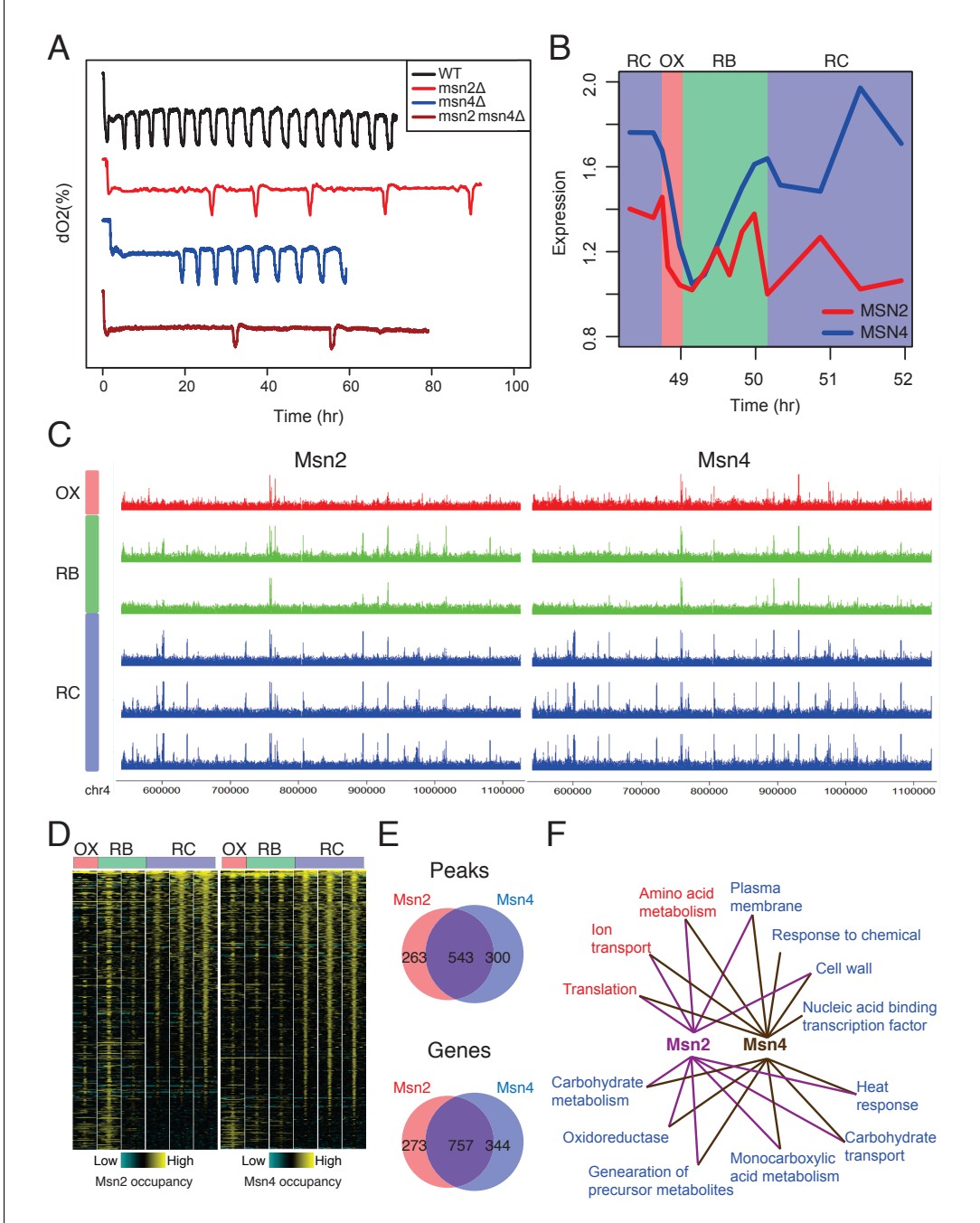

**Figure 2.** Msn2 and Msn4 regulate the length of RC/quiescence phase. (A) Oscillation defects in msn2Δ, msn4Δ and msn2Δmsn4Δ mutants. Similar data are presented for completeness in the companion paper by *Kuang et al. (2017) Figure 2C*. (B) Dynamics of MSN2 and MSN4 transcripts in YMC. (C) A snapshot of Msn2 and Msn4 binding across the six time points in YMC. Tracks represent ChIP-seq signals at consecutive time points. Note the consistency of pattern during the RC phase. (D) Temporal patterns of Msn2/4 binding centered at each binding site and flanked by 1000 bp. (E) Venn graphs showing substantial overlaps between Msn2 and Msn4 peaks and target genes. (F) Enriched GO terms of Msn2 and Msn4 target genes. Red are terms of OX phase genes and blue are terms of RC phase genes. RB phase genes did not have statistically significant GO Term enrichment.
DOI: https://doi.org/10.7554/eLife.29938.006

The following source data and figure supplement are available for figure 2:

**Source data 1.** Binding sites of Msn2.
DOI: https://doi.org/10.7554/eLife.29938.008
**Source data 2.** Binding sites of Msn4.
*Figure 2 continued on next page*

Figure 2 continued

DOI: https://doi.org/10.7554/eLife.29938.009

**Figure supplement 1.** Western blots of WT, *msn2Δ* and *msn4Δ* lysates against Msn2 and Msn4 antibodies.
DOI: https://doi.org/10.7554/eLife.29938.007

phase to represent the relatively longer time of the RC phases (*Figure 3B*). We measured acetyl-CoA levels in WT and mutant cells and observed a dramatic delay of acetyl-CoA accumulation in the mutant cells (*Figure 3C*). Together, the results suggested that Msn2/4 control the transition from RC to OX phase by regulating the accumulation of acetyl-CoA.

We further asked how Msn2/4 regulate the acetyl-CoA level. Glycolysis and fatty acid oxidation are the major carbon metabolic pathways activated in the RC phase (*Figure 1A*) (*Tu et al., 2005*). DynaMO prediction analysis indicated that 20 of the 33 genes encoding glycolytic enzymes were predicted to be bound by Msn2/4 (*Figure 3D*) and the number of predicted binding sites was significantly higher than the number of sites from random sets of genes ($p<1\times10^{-5}$) (*Figure 3—figure supplement 2E*). We confirmed the observation by examining the ChIP-seq results. 27 out of the 33 genes were bound by Msn2/4 (*Figure 3D* and *Figure 3—figure supplement 1A*) and intriguingly, binding of Msn2/4 extends into the coding regions for some of the genes, which may suggest some unknown regulatory mechanisms. The coding regions of glycolytic genes are more conserved than the coding regions of other genes and the promoter regions of glycolytic genes in *Saccaromyces* strains (*Figure 3—figure supplement 1B*). The number of motif sites overlapping Msn2/4 peaks was significantly higher than the number of sites from random sets of 33 genes ($p<1\times10^{-7}$; *Figure 3—figure supplement 2C,F*). On the other hand, only 2 of 11 genes encoding enzymes in fatty acid oxidation were bound by Msn2/4 (*Figure 3—figure supplement 2A*). To further challenge the hypothesis that Msn2/4 regulate the level of acetyl-CoA through glycolysis, we examined the dynamics of mRNA levels of glycolytic genes in WT and *msn2Δmsn4Δ* double mutant by RT-qPCR. These genes were highly induced in the WT cells during the RC phase but were surprisingly dramatically reduced in the mutant (*Figure 3E*). Collectively, the results suggest that Msn2 and Msn4 control the transition from RC/quiescent state to OX/growth state by regulating the intracellular level of acetyl-CoA through glycolysis. Although we lack direct evidence for which intracellular pool of acetyl-CoA is regulated by Msn2/4, the pyruvate dehydrogenase (PDH) complex, which functions in mitochondria, was not targeted by Msn2/4, whereas multiple cytosolic enzymes were targeted by Msn2/4 (*Figure 3D*), suggesting that the nucleo-cytosolic pool of acetyl-CoA might be specifically regulated. It is also consistent with two other lines of previously obtained evidence: (1) that acetate, ethanol and acetaldehyde can induce the transition from RC to OX and (2) that histone acetylation is dramatically increased during such transitions (*Cai et al., 2011*). Interestingly, the oncogene *MYC* similarly regulates glycolytic genes in mammals (*Dang, 2012*).

Additionally, we asked how Msn2/4 affect the initiation of the cellular growth program from quiescence when we added 6 day stationary phase cells into fresh YPD medium. Glucose is at 2% in fresh YPD medium, which is much higher than the concentration in the continuous culture of the YMC. Interestingly, we still observed a delay of growth in the *msn2Δmsn4Δ* double mutant compared to the WT strain (*Figure 3—figure supplement 3A*). Inoculation in YP +2% galactose exhibited a bigger delay, probably because Msn2/4 also control the expression of genes in the galactose metabolism pathway (*Figure 3—figure supplement 3B*). The *msn2Δmsn4Δ* double mutant also showed a much lower saturation titer compared to the WT (*Figure 3—figure supplement 3A*). Deletion of *MSN2* and *MSN4* did not affect survival rate in glucose medium (*Figure 3—figure supplement 3B*) but decreased the size of stationary cells in glucose medium (*Figure 3—figure supplement 3C*).

## Regulatory network for Msn2/4 in YMC

To further characterize the functions of Msn2/4 in the YMC, particularly in the lengthy mutant RC/quiescence phase, we performed RNA-seq of 7 time points for WT cycling cells and 11 time points for *msn2Δmsn4Δ* double mutant cells across one cycle (*Figure 3B*). We first compared expression patterns of cycling genes in WT and mutant cells (*Figure 4A* and *Figure 4—source data 1*). The three clusters of gene expression patterns were still observed in the mutant cells. However, many

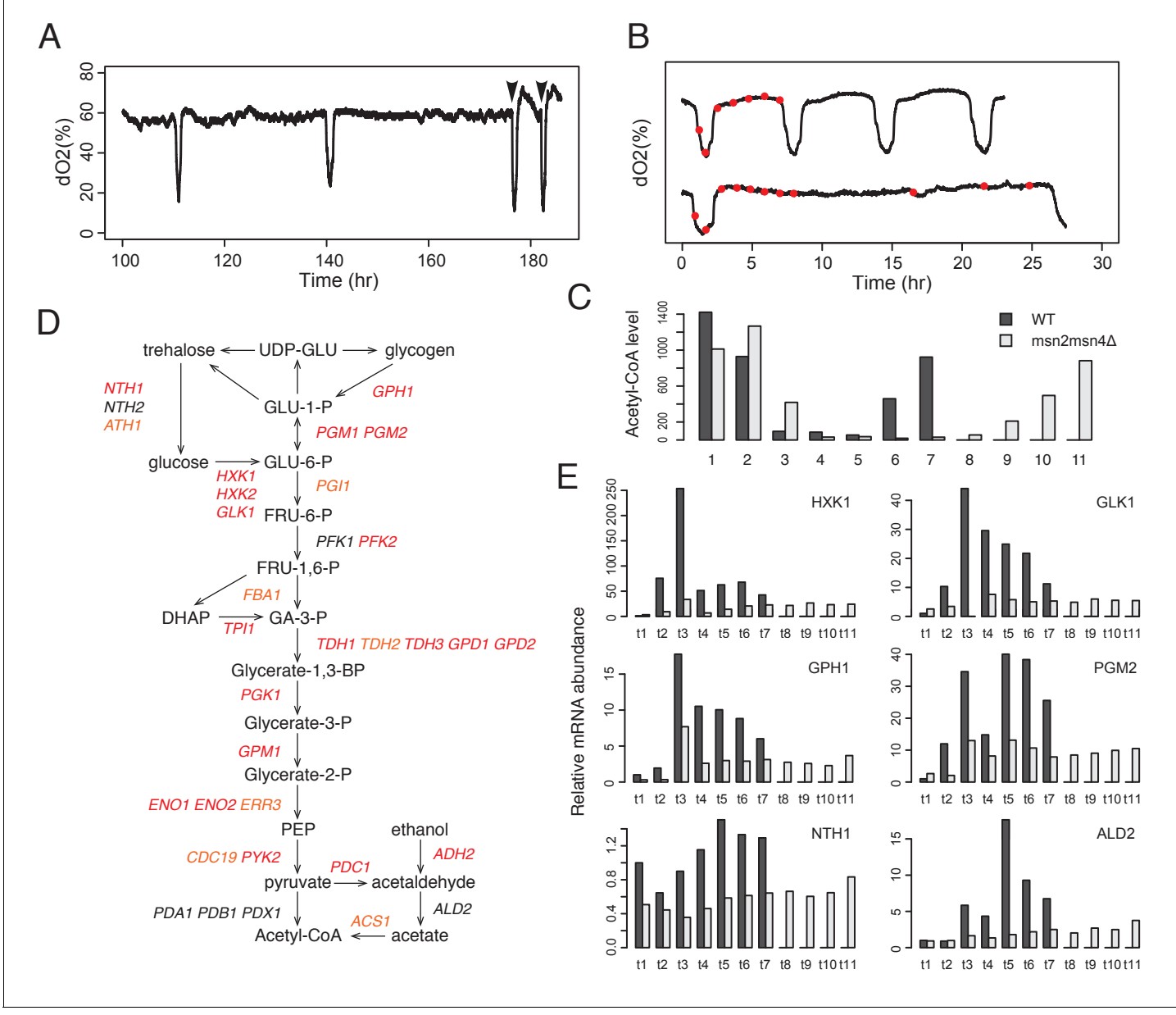

**Figure 3.** Msn2 and Msn4 regulate accumulation of acetyl-CoA through glycolysis. (A) Acetate addition immediately induces OX phase. The graph shows the dO2 oscillation of msn2Δ and the arrows mark the time when acetate was added. (B) Seven time points of WT cells and eleven time points of msn2Δmsn4Δ mutant cells were collected for examination of acetyl-CoA level and gene expression level. The first seven time points of mutant cells have the same absolute time intervals as the seven time points of WT cells. The remaining four time points of mutant cells represent the relative positions in the RC phase of WT cells. (C) shows the relative acetyl-CoA levels in one cycle of WT or msn2Δmsn4Δ cells. (D) shows the glycolysis pathway and genes encoding the enzymes at each step. Red genes are predicted to be bound by Msn2/4 using DynaMO and subsequently validated by ChIP-seq. Orange genes are not predicted but actually bound by Msn2/4. Black genes are neither predicted nor validated to be bound by Msn2/4. (E) shows the RT-PCR results of representative glycolytic genes in one cycle of WT or msn2Δmsn4Δ cells.

DOI: https://doi.org/10.7554/eLife.29938.010

The following figure supplements are available for figure 3:

**Figure supplement 1.** Msn2/4 binding signals at glycolytic genes.

DOI: https://doi.org/10.7554/eLife.29938.011

**Figure supplement 2.** Combinatorial analysis of Msn2 and Msn4 motifs and ChIP-seq in YMC.

DOI: https://doi.org/10.7554/eLife.29938.012

**Figure supplement 3.** Msn2/4 regulate the transition from quiescence to growth in batch culture.

DOI: https://doi.org/10.7554/eLife.29938.013

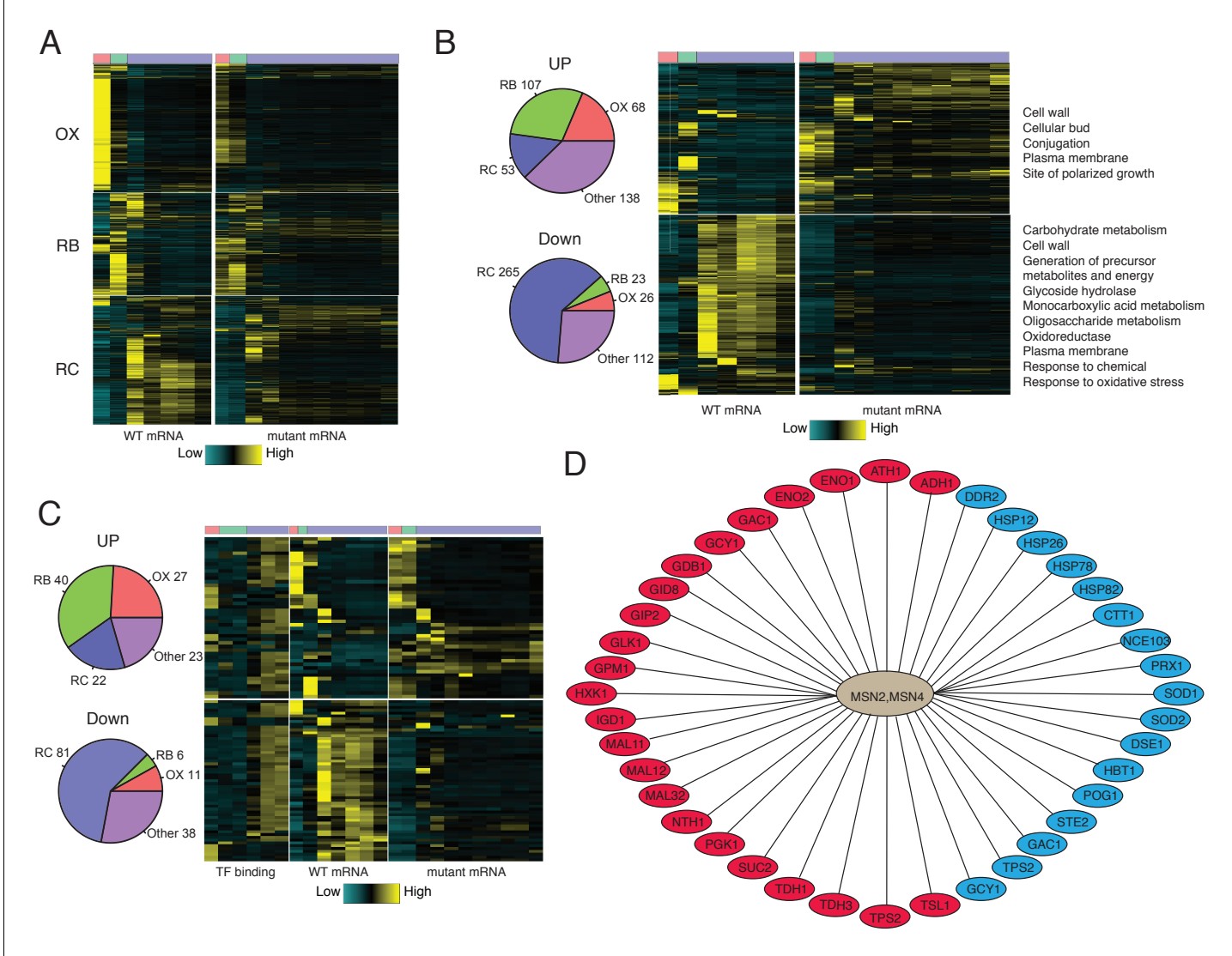

**Figure 4.** Analysis of Msn2/4 targets by RNA-seq and ChIP-seq. (**A**) A heat-map shows temporal expression patterns of all OX, RB and RC phase genes in WT and msn2Δmsn4Δ cycling cells. More samples were taken of the mutant cells due to their elongated YMC. (**B**) A heatmap shows temporal expression patterns of differential expressed genes between WT and msn2Δmsn4Δ cycling cells. Pie charts show the numbers of up- or down-regulated genes in OX, RB, RC and non-cycling (other) genes. (**C**) Temporal expression patterns and Msn2/4 binding patterns of 'core' Msn2/4 target genes. 'Core' targets were genes bound by Msn2/4 and expressed differentially in msn2Δmsn4Δ cells. Pie charts show the numbers of core targets in OX, RB, RC and non-cycling (other) genes. (**D**) Representative 'core' Msn2/4 target genes. Genes in red ovals are carbohydrate metabolism genes and those in blue ovals are genes encoding stress responses.

DOI: https://doi.org/10.7554/eLife.29938.014

The following source data and figure supplement are available for figure 4:

**Source data 1.** FPKMs for seven time point WT and eleven time point *msn2Δmsn4Δ* double mutant across YMC.
DOI: https://doi.org/10.7554/eLife.29938.016
**Source data 2.** Msn2/4 targets characterized by Msn2/4 binding and expression levels of target genes in the *msn2Δmsn4Δ* mutant.
DOI: https://doi.org/10.7554/eLife.29938.017
**Source data 3.** Stress related and mitochondrion related genes in YMC.
DOI: https://doi.org/10.7554/eLife.29938.018
**Figure supplement 1.** Comparison of RC phase cells with nutrient limitation or starvation quiescence cells.
DOI: https://doi.org/10.7554/eLife.29938.015

RC genes were clearly down-regulated. Next, we identified 366 genes up-regulated and 426 genes down-regulated in mutant compared to WT cells at comparable time points. Whereas the 366 up-regulated genes were relatively equally distributed among OX, RB, RC and non-cycling groups, the majority (62.2%) of the 426 down-regulated genes were RC phase genes (*Figure 4B* and *Figure 4— source data 2*). For both up-regulated and down-regulated genes, the changes were primarily observed in the RC phase (*Figure 4B*). In the subsequent analyses, we focused on the down-regulated genes. Among these genes, we found significant enrichment of genes related to carbohydrate metabolism and response to various stresses (*Figure 4B*), and the GO terms were very similar to those detected by ChIP-seq analysis (*Figure 2F*). *FAA1* and *POX1*, which are directly involved in fatty acid β-oxidation and bound by Msn2/4, did not show significant changes of mRNA level in the mutant cells (*Figure 4—source data 2*). This argues against the hypothesis that Msn2/4 promote acetyl-CoA production through fatty acid oxidation.

We next identified 'core' Msn2/4 target genes that were both bound by Msn2/4 and showed differential expression when comparing *msn2Δmsn4Δ* double mutant with WT cells. 112 genes, including 27 OX, 40 RB, 22 RC and 23 non-cycling genes, appear to be repressed by Msn2/4 in WT cells (*Figure 4C*; transcripts are up in mutant cells). 136 genes appear to be directly activated by Msn2/4 in WT cells (i.e., Down in mutant cells) and more than half of them (81 genes) were RC phase genes (*Figure 4C*). The timing of Msn2/4 binding is consistent with their activating function for target RC phase genes (*Figure 4C*). Carbohydrate metabolism genes and stress response genes are the two major classes in the core Msn2/4 targets (*Figure 4D*). As shown previously, many genes encoding glycolytic enzymes, such as *HXK1, GLK1, ENO1, ENO2, PGK1, GPM1, TDH1* and *TDH3* are core Msn2/4 targets. Heat-shock protein genes such as *HSP12, HSP26, HSP78 and HSP82*, oxidative stress response genes including *CTT1, PRX1, SOD1, SOD2* are also core Msn2/4 targets. Certain metabolic genes are also already annotated as stress response genes. For example, *TPS2* and *GAC1*, which are involved in the synthesis of trehalose and glycogen, two storage carbohydrates, are important for stress response. Interestingly, trehalose and glycogen have also been shown to be metabolized during the exit of quiescence to fuel the regrowth (*Shi et al., 2010*; *Silljé et al., 1999*). Msn2/4 may also facilitate these processes by activating *NTH1, ATH1, GPH1, PGM1* and *PGM2* (*Figure 3D*). Therefore, the regulatory network for Msn2/4 suggests parallel mechanisms for how yeast responds to limited nutrients. Stress response genes are activated to maintain survival by removing detrimental factors like mis-folded proteins and oxygen radicals and improving defenses. Carbohydrate metabolism genes are also activated so that cells can utilize the very limited nutrients in the environment for growth, as may often be the case in recovery from quiescence.

## Systematic comparison between YMC and distinct quiescent states

Given that Msn2/4 are involved in the transition from quiescence to growth, we next attempted to characterize the quiescent (RC) state of YMC by comparing it with multiple previously defined quiescent states, including starvation and limitation of one of the three essential nutrients, glucose, nitrogen or phosphate (*Klosinska et al., 2011*). We examined the transcription of cycling genes in the YMC and nutrient-scarce conditions (*Figure 4—figure supplement 1A*). The majority of the OX/ growth genes, which function in biomass synthesis, are turned off when nutrients are scarce, as are the cell cycle genes which peak in the RB/proliferation phase. Interestingly, mitochondrial genes, which also peak in the RB phase, are relatively elevated upon nutrient scarcity. Genes encoding mitochondrial ribosome components are induced temporally at the beginning of glucose starvation or phosphate limitation but not under other conditions while genes encoding translation factors are activated modestly across all nutrient scarce conditions (*Figure 4—figure supplement 1B*). It will be interesting to explore the function of mitochondrial genes when nutrients are scarce. Not surprisingly, most RC genes are induced under nutrient scarce conditions too, reflecting the similarity between the RC phase and other quiescent states. We specifically examined genes encoding glycolytic enzymes, most of which are expressed in the RC phase (*Figure 4—figure supplement 1C*). The majority of these genes are also turned on under nutrient starvation or limitation conditions, suggesting a signal importance of in quiescent states. Details of the comparison are in the supplemental information.

## Discussion

In this study, we applied DynaMO analysis to the YMC, which led us to explore the function of two TFs, Msn2 and Msn4, in regulating the transition from a RC/quiescent state to an OX/growth state. Most previous studies on Msn2/4 were performed under various stress conditions such as starvation, heat shock, osmotic and oxidative stresses and used cell viability as readout (*Estruch and Carlson, 1993*; *Estruch, 2000*; *Martínez-Pastor et al., 1996*; *Schmitt and McEntee, 1996*). Although colony formation can be seen as a single round of regrowth, many details such as regrowth dynamics were ignored in such assays. In the YMC, in which nutrient is continuously limited, growth, proliferation and quiescence are temporally separated, providing a platform for mechanistic exploration of transitions between distinct cellular states. The unusual 'lengthened RC phase' phenotype in *MSN2* and *MSN4* mutants is consistent with a transcriptional program in the quiescent phase which could be critical for subsequent cell proliferation. Our analyses suggest that Msn2/4 modulate the cellular programs by changing the intracellular acetyl-CoA levels, probably by promoting glycolysis. Metabolism has strong effects on cell growth and proliferation. Metabolites such as acetyl-CoA have been shown to function not only as building blocks for the synthesis of fatty acids, amino acids and nucleotides, but also as signals for gene expression and enzymatic activities (*Lyssiotis and Cantley, 2014*). Acetyl-CoA has been proposed as a central metabolite that regulates cellular growth program from yeast to cancer cells (*Cai et al., 2011*; *Comerford et al., 2014*; *Mashimo et al., 2014*). Does a transcriptional program exist to control the acetyl-CoA level? Our analyses offer evidence that in budding yeast, transcription factors Msn2 and Msn4, well known as stress response factors, function as key transcription factors that activate genes encoding glycolytic enzymes. By controlling expression of these enzymes, Msn2/4 may support the generation of acetyl-CoA in preparation for cell growth and proliferation. This regulatory mode is not obvious when cells are grown in rich medium, probably because acetyl-CoA level is high enough to support continuous growth. But it becomes extremely important when cells are under nutrient-limited condition. Supposedly, nutrient limitation is sensed by TOR signaling and inhibition of TORC1 leads to nuclear translocation and activation of Msn2/4 and other stress response TFs (*Beck and Hall, 1999*; *Loewith and Hall, 2011*). That probably explains why many stress response TFs, such as Gis1, Mig1/2/3, and Msn2/4, that target metabolic enzymes were identified in the RC/quiescent phase of YMC by DynaMO (*Figure 1*). Therefore, this study seemingly uncover a critically important secondary mechanism for stress response – in the quiescent state, remodeling metabolic activities and activating stress responses and defenses pathways to maintain survival, and then enabling rapidly regrowth once nutrients are replenished (*Ho and Gasch, 2015*). It will be interesting to examine whether deletion of any of these TFs show similar effects on YMC and whether and how these TFs collaborate with each other in regulating the key metabolic genes and metabolites for recovery from quiescence.

Msn2/4-dependent glycolysis potentially provides fuels supporting cell growth under continuous nutrient limited conditions, similar to the situation in cancers, which are also associated with robust glycolysis and survive under nutrient limited conditions (*Comerford et al., 2014*). One speculative hypothesis is that a functional equivalent of Msn2/4 in mammalian cancer is *MYC*, which similarly targets all glycolytic enzymes and fosters tumor growth (*Dang, 2012*). It suggests that TF dependent activation of glycolysis to support cell proliferation under nutrient limited condition can be a prevalent biological motif.

## Materials and methods

### Metabolic cycles

Metabolic cycle experiments were performed as previously described (*Kuang et al., 2014*). A BioFlo 3000 from New Brunswick Scientific was used. YMC runs were operated at an agitation speed of 475 rpm (Bioflo 3000), an aeration rate of 1 L/min, a temperature of 30°C, and a pH of 3.4 in 1 L YMC medium. After the batch culture was saturated for at least 4 hr, fresh medium was added continuously at a dilution rate of $\sim$0.09 $\sim$0.1 h$^{-1}$. three independent isolates of each mutant were tested for YMC and representative curves were presented in *Figures 2* and *3*.

## Strains and media

YP + glucose or galactose medium contains 1% yeast extract, 2% bacto-peptone, 2% dextrose or galactose and 1.6 mM tryptophan. 200 µg/ml G418 or 100 µg/ml ClonNat or 300 µg/ml Hygromycin were supplemented in YPD for drug resistance selection. The YMC medium consists of 5 g/L $(NH_4)_2SO_4$, 2 g/L $KH_2PO_4$, 0.5 g/L $MgSO_4 \bullet 7H_2O$, 0.1 g/L $CaCl_2 \bullet 2H_2O$, 0.02 g/L $FeSO_4 \bullet 7H_2O$, 0.01 g/L $ZnSO_4 \bullet 7H_2O$, 0.005 g/L $CuSO_4 \bullet 5H_2O$, 0.001 g/L $MnCl_2 \bullet 4H_2O$, 1 g/L yeast extract, 10 g/L glucose, 0.5 mL/L 70% $H_2SO_4$, and 0.5 mL/L Antifoam 204 (Sigma) (*Kuang et al., 2014*).

All strains were generated from the CEN.PK background and manipulated by standard budding yeast protocols:

Yeast strains:

| Name | Background | Genotype |
| --- | --- | --- |
| BY5764 | CEN.PK | *MATa* |
| ZKY749 | CEN.PK | *MATa msn2Δ::KanMX6* |
| ZKY750 | CEN.PK | *MATa msn4Δ::KanMX6* |
| ZKY756 | CEN.PK | *MATa msn2Δ::hygMX, msn4Δ::KanMX6* |

Gene knockout strains were generated by homologous recombination using PCR products containing a drug cassette (*kanMX6* or *hygMX*) and 40 bp sequences flanking the target gene. Tagged-protein strains were generated similarly by integrating a cassette containing a protein tag and a drug resistance cassette at C terminus (*Kuang et al., 2014*). PCR products were transformed into a diploid strain and the heterozygous diploids were sporulated and dissected to select for haploids with drug resistance.

## RT-qPCR and RNA-seq

2 OD cycling cells were collected and flash frozen. RNA was extracted with the Qiagen RNeasy Mini kit (QIAGEN, 74104, Valencia, CA). First strand cDNA was synthesized by reverse-transcription using oligo(dT)$_{20}$ primer from SuperScript III First-Strand Synthesis System (Invitrogen, 18080–051, Grand Island, NY). Fast SYBR Green Master Mix (Applied Biosystems, 4385612, Foster City, CA) was used for real-time PCR and experiments were done on the platform of StepOnePlus Real-Time PCR System (Applied Biosystems, 4385612, Foster City, CA). RNA-seq libraries were prepared in the New York University Genome Technology Center using Illumina Trueseq RNAseq v2 library kit (Illumina, San Diego, CA). PolyA beads were used for mRNA selection. 500 ng of RNA per sample was used as input and 12 cycles of PCR were run for amplification. Libraries were pooled together and sequencing was performed on Hiseq platform.

## ChIP-seq

ChIP was performed as previously described (*Kuang et al., 2014*).~50 OD WT cycling cells per time point were collected for ChIP of Msn2 and Msn4. 6 time points were used to represent all three phases of YMC and they were relatively evenly distributed across the cycle. Antibodies are as following: Msn2 (y-300, sc-33631, RRID:AB_672215), Msn4 (yE-19, sc-15550, RRID:AB_672217). Validation is provided on the manufacturer's website and the antibodies were further tested by western blots of *WT*, *msn2Δ* and *msn4Δ* lysates (*Figure 2—figure supplement 1*). 2.5 µg primary antibody was used per ChIP experiment. Briefly, cells were first fixed in 1% formaldehyde at 25°C for 15 min and quenched in 125 mM glycine at 25°C for 10 min. Cells were pelleted and washed twice with TBS buffer before freezing. The frozen pellet was resuspended in 0.5 ml ChIP lysis buffer (50 mM HEPES•KOH pH 7.5, 500 mM NaCl, 1 mM EDTA, 1% Triton X-100, 0.1% deoxycholate (DOC), 0.1% SDS, 1 mM PMSF, 5 µM pepstatin A, Roche protease inhibitor cocktail) and split into two tubes and lysed by bead beating. Lysate were combined and expanded into 1 ml and sonicated for 16 cycles (30 s on, 1 min off, high output) using a Bioruptor (Diagenode, Denville, NJ). The supernatant of the sonicated lysate was pre-cleared and incubated with 2.5 µg primary antibodies. After incubation overnight, 50 µl protein G magnetic beads (Invitrogen, Grand Island, NY, 10003D) were added and incubated for 1.5 hr at 4°C. Beads were washed twice with ChIP lysis buffer, twice with DOC buffer (10 mM Tris•Cl pH 8.0, 0.25 M LiCl, 0.5% deoxycholate, 0.5% NP-40, 1 mM EDTA) and twice with

TE. 100 µl of TES buffer (TE pH8.0 with 1% SDS, 150 mM NaCl, and 5 mM dithiothreitol) was added to resuspend the beads at 65°C for 20 min. Reverse crosslinking was performed by incubation for 6 hr at 65°C. An equal volume of TE containing 1.25 mg/ml proteinase K and 0.4 mg/ml glycogen was added to the samples after reverse crosslinking and samples were incubated for 3 hr at 37°C. DNA samples were purified using ChIP DNA Clean and Concentrator (ZYMO RESEARCH, D5205, Irvine, CA). Library construction and sequencing were performed using KAPA Hyper Prep Kit (KAPABIO-SYSTEMS, KK8502, Wilmington, MA). Briefly, DNA was end repaired and A-tailed. Barcoded adaptors were ligated and DNA was purified with Agencourt AMPure XP beads (Beckman Coulter, A63880, Indianapolis, IN) and amplified by 12–16 cycles. PCR products were gel-extracted and quantified on an Agilent Bioanalyzer. Sequencing was performed on Illumina Hiseq platform. Raw reads were mapped to the reference genome (sacCer2) by bowtie (*Langmead et al., 2009*) and peaks were visualized by the CisGenome Browser (*Ji et al., 2008*).

## Acetyl-CoA measurement

Acetyl-CoA was extracted with two methods. For the first method, Na azide was added (10 mM) and 5 OD cells were spun down and lysed in 200 µl of 10% perchloric acid by bead beating. The lysate was spun down and the supernatant was neutralized to pH 6–8 with 3 M K bicarbonate, with vortexing and cooling on ice for 5 min. K bicarbonate was spun down and supernatants were used. For the second method, 5 OD cells were resuspended in 4 mL Quenching Solution M (60% methanol, 10 mM Tricine pH7.4) and incubated at −40°C for 5 min. The cells were spin at 1000 g for 3 min at −10°C and washed once with Quenching Solution M. Pellets were resuspended in 1 mL Extraction Buffer M (75% ethanol, 0.5 mM Tricine pH 7.4) and incubated at 80°C for 3 min. Mix was cooled on ice for 5 min and spun down. Supernatants were dried down in speedvac and dissolved in 30 µl water. The concentration of acetyl-CoA was measured using Acetyl-Coenzyme A Assay Kit (Sigma, MAK039, St. Louis, MO).

## Yeast survival measurements

Cells in log or stationary phase were counted microscopically in a hemocytometer. A fixed number of cells was plated on YPD plates and the colonies were counted. The survival rate was the number of colonies/number of cells plated. Three biological replicates were examined.

## Cell size measurement

Log or stationary phase cells were harvested and washed with PBS twice. Forward scatter (FSC) was measured by BD Accuri C6 Flow Cytometer (BD Biosciences, Franklin Lakes, NJ) and used as the indicator of cell size. Two biological replicated were examined and plotted.

## DynaMO

The DynaMO algorithm is described in detail in the manuscript (*Kuang et al., 2017*).

## Additional data analyses in the YMC

### Msn2/Msn4 ChIP-seq data analysis

To evaluate the overlap between Msn2 and Msn4 binding sites (*Figure 2*), Msn2 and Msn4 peaks were combined across six time points using the 'reduce' function from the 'GenomicRanges' package. We then counted the numbers of overlapping Msn2 and Msn4 total or time-specific peaks and displayed in venn diagrams. To identify Msn2 and Msn4 target genes, we took the region of a gene from 700 bp upstream of the start codon to the stop codon and examined if the gene region overlapped a Msn2 or Msn4 peak. These genes were candidate targets. Candidate targets that are differentially expressed between WT and msn2msn4 mutant are declared as core target genes (see below). Gene ontology analysis was performed based on the SGD (RRID:SCR_004694) annotation file (http://www.yeastgenome.org/download-data/curation#.UQFqKhyLFXY) with Fisher exact test. P values were converted into FDR by p.adjust to adjust for multiple testing.

### RNA-seq data analysis

RNA-seq reads were mapped against SacCer2 genome using bowtie (*Langmead et al., 2009*). Read counts per gene was measured in R using the 'CountOverlaps' function. Differential expressed genes

were identified using DESeq (*Anders and Huber, 2010*) package. We first compared pairs of WT mutant samples as follows: WT1-Mut1, WT2-Mut2, WT3-Mut3, WT4-Mut4, WT5-Mut5, WT6-Mut6, WT7-Mut7, WT4-Mut8, WT5-Mut9, WT6-Mut10, WT7-Mut11. In the first seven comparisons, we matched samples at the same absolute time in YMC. In the next four comparisons, we matched samples at the same relative time in the RC phase of YMC. For each pair, genes with adjusted p value less than 0.01 were identified as differentially expressed genes. We then compared the 5 WT samples in RC phase to the nine mutant samples in RC phase. Differentiated expressed genes from all above comparisons were grouped together as up- and down- regulated genes. Genes that were bound by Msn2/4 and showed differential expression in mutant cells were defined as 'core' targets. GO analysis was performed same as above.

## Motif site enrichment analysis

To examine the enrichment of Msn2 and Msn4 motif sites in different processes, yeast genes were extended by 1000 bp upstream from the start codon. Glycolysis, fatty acid oxidation and galactose metabolism genes were selected based on SGD annotation and previous study (*Tu et al., 2005*). We examined total motif sites (T), predicted binding sites (Predicted) and motif sites overlapping TF ChIP-seq binding peaks. Numbers of total motif sites, predicted and ChIP-seq identified TF-bound motif sites were calculated in the gene regions of interested pathways and scrambled gene sets. To get scrambled gene sets, we randomly chose the same number of genes from the genome 100 times and examined if motif sites were located within the regions of these genes. To evaluate the distribution of Msn2 and Msn4 motif sites in human glycolytic genes, we first detected motif sites in hg19 genome using the yeast Msn2/4 motif sequences with CisGenome. We then identified the human homologs of yeast glycolysis genes using the BioMart tool from Ensembl and extended the gene regions by 20,000 bp upstream from the transcription start sites. Similarly, total motif sites or those overlapping with DNase I hypersensitive regions were examined in these genes and random selected genes. The human DNase-seq data for 57 cell types were downloaded from the ENCODE project through http://hgdownload.cse.ucsc.edu/goldenPath/hg19/encodeDCC/wgEncodeUwDnase. Then, the whole genome (chromosome Y excluded) was separated by 200 base pair bins (i.e. genomic loci). Bins with read counts larger than 10,000 in one or more cell types (abnormal loci) and bins with read counts smaller than 10 in all cell types (noisy loci) were excluded. After filtering, 1,108,603 genomic loci with unambiguous DNase-seq signal in at least one cell type were retained.

## Conservation analysis

PhastCons scores for multiple alignments of the yeast strains to the Saccaromyces cerevisiae genome are downloaded from http://hgdownload.cse.ucsc.edu/goldenPath/sacCer2/phastCons7-way/. Six strains were used for alignment, *Saccharomyces paradoxus*, *Saccharomyces mikatae*, *Saccharomyces kudriavzevii*, *Saccharomyces bayanus*, *Saccharomyces castelli*, and *Saccharomyces kluyveri*. Promoter regions were calculated as 0–500 bp upstream of start codons and ORF regions were regions from start codons to stop codons. Scores were averaged across each base pair and further averaged across interesting groups of genes.

## Comparison between RC phase with other quiescence phases

Data of gene expression, nutrient-specific genes and genes required for specific nutrient starvation were downloaded from a previous study (*Klosinska et al., 2011*). Gene expression data in nutrient quiescence conditions and YMC were merged and displayed in a heat map. Mitochondrial genes in the RB phase were selected by intersecting RB phase genes and gene annotated as 'Mitochondrion' by SGD. Numbers of nutrient-specific genes and genes required for nutrient starvation intersected with OX, RB and RC phase genes were counted and enrichment levels were evaluated by Fisher exact tests. Adjusted p values were displayed in heatmap. Expression levels of nutrient-specific genes in YMC were examined by evaluating the max FPKMs across 16 time points of YMC. Msn2/4 target genes or randomly selected genes were intersected with nutrient-specific genes or genes required for nutrient starvation.

## Data accession

RNA-seq and ChIP-seq data have been deposited in the Gene Expression Omnibus database under accession code GSE72263 (https://www.ncbi.nlm.nih.gov/geo/query/acc.cgi?acc=GSE72263).

## Acknowledgements

We thank B Tu and L Shi for suggestions on YMC. We thank Mordechai Choder for discussions on metabolic cycle and transcriptional control. We thank Adriana Heguy and her staff at the New York University School of Medicine Genome Technology Center for expert assistance with RNA-Seq and ChIP-Seq. This work was supported by the Technology Center for Networks and Pathways grant U54GM103520 and the research grant R01HG006841 and R01HG006282 from the NIH to JDB and HJ.

## Additional information

### Funding

| Funder | Grant reference number | Author |
| --- | --- | --- |
| National Institutes of Health | U54GM103520 | Zheng Kuang<br>Jef D Boeke |
| National Institutes of Health | R01HG006841 | Zheng Kuang<br>Hongkai Ji |
| National Institutes of Health | R01HG006282 | Zheng Kuang<br>Hongkai Ji |

The funders had no role in study design, data collection and interpretation, or the decision to submit the work for publication.

### Author contributions

Zheng Kuang, Conceptualization, Resources, Data curation, Software, Formal analysis, Validation, Investigation, Methodology, Writing—original draft, Project administration; Sudarshan Pinglay, Data curation, Validation, Investigation, Writing—review and editing; Hongkai Ji, Conceptualization, Resources, Software, Supervision, Funding acquisition, Methodology, Writing—review and editing; Jef D Boeke, Conceptualization, Resources, Supervision, Funding acquisition, Project administration, Writing—review and editing

### Author ORCIDs

Zheng Kuang http://orcid.org/0000-0001-5855-8371
Sudarshan Pinglay https://orcid.org/0000-0002-8781-1476
Jef D Boeke https://orcid.org/0000-0001-5322-4946

### Decision letter and Author response

Decision letter https://doi.org/10.7554/eLife.29938.027
Author response https://doi.org/10.7554/eLife.29938.028

## Additional files

### Supplementary files

• Transparent reporting form
DOI: https://doi.org/10.7554/eLife.29938.020

### Major datasets

The following dataset was generated:

| Author(s) | Year | Dataset title | Dataset URL | Database, license, and accessibility information |
|---|---|---|---|---|
| Kuang Z, Boeke JD, Ji H | 2017 | DynaMO, a package identifying transcription factor binding sites in dynamical ChIPSeq/RNASeq datasets, identifies transcription factors driving yeast ultradian and mammalian circadian cycles | https://www.ncbi.nlm.nih.gov/geo/query/acc.cgi?acc=GSE72263 | Publicly available at the NCBI Gene Expression Omnibus (accession no: GSE72263) |

The following previously published dataset was used:

| Author(s) | Year | Dataset title | Dataset URL | Database, license, and accessibility information |
|---|---|---|---|---|
| Kuang Z, Cai L, Zhang X, Ji H, Tu BP, Boeke JD | 2014 | A high-resolution view of transcription and chromatin states across distinct metabolic states in budding yeast | https://www.ncbi.nlm.nih.gov/geo/query/acc.cgi?acc=GSE52339 | Publicly available at the NCBI Gene Expression Omnibus (accession no: GSE52339) |

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
