## [Decision Letter]

[Editors’ note: a previous version of this study was rejected after peer review, but the authors submitted for reconsideration. The first decision letter after peer review is shown below.]

Thank you for choosing to send your work, "Stress-response factors Msn2/4 control re-entry of quiescent cells into growth through glycolysis", for consideration at *eLife*. Your submission has been reviewed by three peer reviewers and the evaluation has been overseen by a Reviewing Editor and a Senior Editor.

This paper analyzes the role of the general stress-response master regulators, MSN2 and MSN4 in controlling the yeast metabolic cycle (YMC). Based on the results, the authors conclude that the two factors regulate key transition in this cycle by regulating the accumulation of acetyl-CoA.

As you will see below, all three reviewers appreciated your study in providing interesting new data concerning the regulation of the metabolic cycle. However, there was also a serious concern about your claim of direct relationship between Msn2/4 regulation and acetyl-CoA pool levels (and production of this metabolite). Two of the reviewers were more supportive of this conjecture, yet even these reviewers requested that you will further validate your claims by analyzing the consequences of MSN2,4 over-expression, either transiently or by using a constitutive system

The third reviewer, who is a metabolic expert, maintained that you should focus your study on the role of Msn2/4 in the shift between fermentation and respiration, and remove the claim about direct relationship between Msn2/4 regulation and acetyl-CoA pools. I would like to emphasize that since this reviewer is an expert in metabolism, we will not be accept the paper unless he/she accepts the arguments and claims presented.

In addition, all three reviewers requested that you better discuss previous studies, and in particular refer to previous work on carbohydrate metabolism during nutrient deprivation.

Reviewer #1:

In this manuscript, Kuang and co-authors showed that the general stress responsive factors Msn2 and Msn4 play an important role in driving yeast metabolic cycle (YMC). They further revealed that it is through controlling the accumulation of acetyl-CoA and the same mechanism might generally contribute to cell growth regulation under prolonged nutrient limiting conditions.

I find this paper interesting in that (1) the effect of *msn2/4* deletion to the progression of YMC is striking! (2) the proposed mechanism is based on careful genome-wide analysis and is convincing. (3) the potential relevance to cell growth regulation in general is intriguing. Therefore, I recommend the acceptance of this manuscript for publication in *eLife* provided the authors are able to address the major points below.

1) The conclusions are based primarily on loss of function mutant phenotypes (*msn2/4* deletion) and the correlation analysis between metabolic enzyme expression or Msn2/4 genomic occupancy with YMC phases. It would significantly strengthen the paper if the authors can show some gain-of-function effects. For example, how does Msn2/4 overexpression affect YMC and the growth transition in general? This is especially important because most people in the field believe that activated Msn2/4 slow down cell growth.

2) Msn2 and Msn4 are activated by dephosphorylation and translocation from the cytoplasm to the nucleus, therefore nuclear localization can serve as a reporter for Msn2/4 activity. In the paper, the authors only monitored the expression changes during YMC. It would serve as the direct supportive evidence for Msn2/4 activation if the authors can observe nuclear localization of Msn2/4 specifically during the RC phase. And if so, what would be the authors' speculation on the mechanisms of Msn2/4 activation? This will provide some clues on why Msn2/4 are activated specifically during the RC phase.

3) The authors claim that Msn2/4 have a dual role in regulating carbohydrate metabolism genes and stress resistance genes. Under the conditions that they are focusing on (prolonged nutrient-limited condition), do the stress resistance genes also contribute to cell growth or they are unrelated by-products? An analysis or at least a discussion on this would be helpful.

Reviewer #2:

This paper describes analysis of how the transcription factors Msn2 and Msn4 regulates expression of genes encoding enzymes of the central carbon metabolism in yeast, with focus on regulation during the metabolic cycle of yeast. These TFs are known to play a central role in stress regulation, and they regulate expression of a very large number of genes in yeast (>100). The authors claim that based on their findings it can be suggested that Msn2/4 control the transition from reductive charging phase to oxidative phase by regulating the accumulation of AcCoA. Regulation of shifts in yeasts metabolism is highly complex, and it tends to the naive to suggest such a statement, and it is by no means supported by the data. Thus, the authors focus their discussion on how Msn2/4 binds (and control expression) to genes of the glycolysis, but these genes are controlled by a large number of other TFs. Furthermore, Msn2/4 themselves control a large number of other genes, including several other TFs, which is not considered in their analysis. A main line of argument is measurements of AcCoA, but this metabolite is present in several cellular compartments – mitochondria, cytosol, peroxisomes and the nucleus (probably equilibrated with the cytosolic pool), and there is no free transport of AcCoA between these compartments. Measuring the total level and linking this to enzymes that are operating in different compartments is not correct (Figure 3 is wrong as ACS1 is forming acetyl-CoA in the cytosol and PDA1 is forming it in the mitochondria). The paper is interesting as the authors present a lot of data, but the biological conclusion they are trying to extract from these data is not supported by the data, and is likely wrong.

Reviewer #3:

In Kuang et al., 2014 the authors examine the role of Msn2 and 4 to regulate glycolytic metabolism in order to promote acetyl-CoA and cell cycle entry. The authors utilize the very elegant YMC system to carefully monitor transitions between reductive and oxidative phases. They first introduce the DynaMO algorithm that identifies transcription factor motifs that influence periodic gene expression in the YMC. Using this method they identify Msn2 and 4, among others, as being involved in the reductive charging phase of the YMC. The authors perform ChIP experiments to demonstrate that Msn2 and 4 chromatin occupancy increases in the RC phase. They further demonstrate that deletion of Msn2/4 causes disruption of the respiration cycles in the YMC to one in which the reductive phase is dramatically lengthened. The authors reason that this lengthened cycle is due to a reduction in acetyl-CoA production. Indeed, ectopic addition of acetyl-CoA can rapidly induce OX phase transition (as has been previously shown by the Tu lab). Moreover, the authors show that RC phase gene expression closely resembles that of nutrient deprivation performed by other labs. Thus, data observed in the YMC is largely applicable to nutrient stress environments in general.

The findings presented in the manuscript are of high significance and broaden the knowledge of factors and mechanisms that influence cell survival. In particular, recent publications on acetyl-CoA demonstrate that this a key metabolite in promoting cell growth and proliferation. This data demonstrates that well-established transcription factors support the metabolism of this critical metabolite. The data in this manuscript is clear and the writing is concise. In general, the conclusions support the main findings of the paper.

The following are specific comments that can improve the manuscript:

1) The authors could do a better job in integrating current literature on the importance of carbohydrate stores, particularly trehalose and glycogen, in survival in nutrient stress conditions and re-entry into the cell cycle from quiescence.

2) Why did the authors focus on H3K9 acetylation, rather than gene expression, in the DynaMO prediction transcription factors that drive the YMC? (This may be detailed in the accompanying manuscript. However, some more details in this manuscript would be helpful to understand the outcome of the results.) Does the focus on H3K9ac provide a better assessment of gene expression that is responsive to acetyl-CoA?

3) The Introduction states that TFs were deleted to validate their role in the YMC. This is a bit vague. Can the authors state which ones? Is Msn2 and 4 unique in terms of its YMC disruption?

4) In the Introduction, the authors state that Msn2 and 4 are known to regulate stress genes. This statement should include relevant references, such as Gasch et al., 2000; Martínez-Pastor et al., 1996; Schmitt and McEntee, 1996).

5) The authors may also want to discuss upstream factors that regulate Msn2 function. In particular Tor signaling, which is influence by nutrient availability.

[Editors’ note: what now follows is the decision letter after the authors submitted for further consideration.]

Thank you for submitting your work entitled "Stress-response factors Msn2/4 control re-entry of quiescent cells into growth through glycolysis" for consideration by *eLife*. Your article has been reviewed by three peer reviewers, and the evaluation has been overseen by a Reviewing Editor and a Senior Editor. The following individuals involved in review of your submission have agreed to reveal their identity: Nan Hao (Reviewer #1); Ashby J Morrison (Reviewer #2).

Our decision has been reached after consultation between the reviewers. Based on these discussions and the individual reviews below, we regret to inform you that your work will not be considered further for publication in *eLife*.

We found the data on Msn2/4 regulating glycolysis convincing and believe it can be published as is. For *eLife*, however, we would have expected more solid insight into how one goes from altered glycolysis to changes in the regulatory molecule acetyl-CoA. This link, unfortunately, is not yet convincing enough and the revision did not help in solidifying this result.

Reviewer #1:

In the revision, the authors did not examine the YMC progression or acetyl-CoA level (which are where the major debates centered on) in the overexpression strain, because "the YMC requires coordination of various parameters"(?). Instead, they just checked the regrowth rate and got a variety of inconsistent results. This is not surprising given that they transformed yeast with a 2 micro plasmid (pRS425) expressing Msn2 or Msn4. The variability of plasmid copy numbers is huge, and therefore the kind of colony variability is expected no matter which output phenotypes (growth rate or others) you will be looking at. A genomic integration (with pRS305) will work way better.

At this point, it seems my attempts to save the debated conclusion did not work out. However, given their striking findings summarized in my first review, I still support the publication of this manuscript, but would suggest the authors to tone down their conclusion and avoid claiming of direct causal or driving relationship.

Reviewer #2:

The authors have addressed my previously stated concerns in the revised manuscript. I believe the manuscript is a significant advancement in our understanding of the Msn2/4 involvement in quiescence to growth transitions and energy metabolism.

Reviewer #3:

The manuscript by Kuang et al., 2014 addresses transcriptional regulation during the yeast metabolic cycle. They attribute a key role of the transcription factors Msn2 and Msn4, well known for their role in the yeast stress response, to regulate the mRNA expression of glycolytic enzymes, during the metabolic cycle in the laboratory yeast strain CenPK (a popular model to synchronize yeast cells in their metabolic cycle). The authors conclude that transcriptional control of glycolytic enzymes though Msn2/Msn4 is causally linked to acetyl-CoA accumulation, triggering re-entry into cell growth.

In principle, the storyline of the manuscript has two parts. The first is the one in which the study authors identify Msn2 and Msn4 as transcriptional regulators of glycolysis during the recovery phase of the metabolic cycle. There may be some little bumps here and there as pointed out by the other reviewers, and these warrant to be addressed. But overall I find this part convincing, and in fact impressive.

Then there is the second part, the authors conclude from these transcriptional changes on how cells regulate metabolism via acetyl-CoA to drive cell growth. This part is not more than a (perhaps reasonable) speculation. It is intrinsically difficult to predict a change in the metabolome from transcriptional data, certainly one cannot conclude from the mRNA level of ~15 glycolytic enzymes on the acetyl-CoA concentration (that is no product of glycolysis) and its role in the recovery phase of the metabolic cycle.

In essence, the title of the manuscript should read something like 'Msn2 / Msn 4 regulate the expression of glycolytic enzymes during the recovery phase of the metabolic cycle'. I'm looking forward to discussing with the other reviewers whether this discovery on its own meets the bar set by *eLife*. I think it perhaps does; *eLife* did aim for very high in the past, but one needs to see this in the light of the recent and effectual situation. One might recommend to the authors however to tone down the manuscript to the part that they actually show.

[Editors’ note: following the rejection, the authors appealed. The appeal was assessed by the editors and revisions were requested prior to acceptance.]

Thanks for asking to re-consider the decision and sorry for the delay in getting back to you. The decision to reject the paper was based primarily on the worry, expressed by one of the reviewers and at least partly agreed to by the others, that, given the inability to establish the functional connection of MSN-related regulation to Acetyl-Co production, the significance of the paper may not reach the desired *eLife* standard.

In particular, please:

1) Tone-down the claims concerning acetyl-coA regulation. All reviewers were of the opinion that your study does not establish a functional connection from transcription regulation of glycolysis to the mechanism leading to altered acetyl-co levels.

2) Validate the antibodies used, as requested by the new reviewer.

3) Please see the other minor comments of the two new reviewers and perhaps emphasize some more the points that they emphasize as being mostly significant about your findings. These revisions are not essential but may improve your presentation.

Reviewer #1

[…] I do believe the authors present some important and novel findings worthy of reconsideration/publication. The data overall look pretty solid to me. It is not well-known that the stress response transcription factors Msn2/Msn4 also regulate glycolytic enzymes, and the evidence looks good that such genes may be induced by Msn2/4 to "prime" cells for rapid sugar utilization and re-entry into growth upon return to favorable nutrient conditions. I believe using the YMC system has enabled them to see this unanticipated relationship between Msn2/4 and glycolytic/carbohydrate metabolic enzymes, which may be obscured in traditional batch culture experiments. I also believe the RNA-seq and ChIP-seq could be high-quality datasets and will be of interest to those investigating both the overlapping and distinct roles for Msn2 and Msn4 in the regulation of stress-responsive gene transcription.

I would ask them to improve Figure 3—figure supplement 1, the legends/labels on the left side of the ChIP-seq views are hard to read, and also comment on the fact that there seems to be binding of these transcription factors throughout the entire coding region for some of these glycolytic genes, in addition to the promoter regions, which could be potentially interesting? Or artifactual? Also, they appear to use commercial antibodies from Santa Cruz for ChIP of Msn2/4, some of their own validation should be included to help demonstrate the specificity of these antibodies (e.g., a Western of WT vs. *msn2∆, msn4∆* lysates).

Reviewer #2

I am sorry about the delay. I had to read this a couple times to appreciate what I think is the importance of this work, but let me emphasize that I have not been up to speed on Ben Tu's most recent work or that of Linda Breeden who has also worked in this area. Hence, there may be things that are more broadly known that were new to me here.

The importance of this paper to my mind is to broaden the appreciation of the roles of Msn2 and Msn4 from general stress resistance transcription factors to regulators of the metabolic cycle that have special roles in metabolism of cells that are coming out of quiescence that are not evident in standard growth conditions. The work is quantitative, incorporating multiple dimensions of analysis (though the magic sauce in DYNAMO was not described) and hence gives us a more comprehensive appreciation of the relevant targets of Msn2 and Msn4. As with any paper that provides a list of genes/proteins, there are some puzzles that are left unanswered such as why GIS1 is a target of this regulation. But on balance, the authors do a good job of drawing together logic behind the interactions described.

In searching for broader relevance, the authors speculate that Myc might be the human equivalent for the regulation of tumor growth since they assert that tumors are nutrient limited. That assertion may be true, but it certainly isn't widely known and raised my skepticism about this point, which is not that important to the results of the paper, but should be checked.

So why did I have to read it a couple times to get it? I think the authors could do a better job of emphasizing how different previous conditions used for the study of Msn2 and Msn4 are from the conditions encountered in the metabolic cycle. I would also advocate losing that damn acronyms for the three phases of the cycle as they are not commonly used and are disruptive to readers who are trying to understand this under explored region of metabolism.

---

## [Author Response]

[Editors’ note: the author responses to the first round of peer review follow.]

Reviewer #1:1) The conclusions are based primarily on loss of function mutant phenotypes (msn2/4 deletion) and the correlation analysis between metabolic enzyme expression or Msn2/4 genomic occupancy with YMC phases. It would significantly strengthen the paper if the authors can show some gain-of-function effects. For example, how does Msn2/4 overexpression affect YMC and the growth transition in general? This is especially important because most people in the field believe that activated Msn2/4 slow down cell growth.

Because the YMC requires coordination of various parameters, it is not ideal for overexpression tests. Instead, we used inoculation of a stationary culture (similar in Figure 3—figure supplement 3) to examine effects of overexpression. We overexpressed MSN2 or MSN4 from 2u plasmid pRS425 in a WT CEN.PK strain and inoculate 6-day stationary cells in YP+0.1% glucose or YP+2% glucose fresh medium. We observed a variety of colony sizes and regrowth phenotypes probably due to the different levels of overexpression and the diverse functions of Msn2/4 target genes (See Author response image 1). Many Msn2/4 target genes show decreased growth rates in both deletion and over-expression mutants, for example HSP12. A medium size colony overexpressing MSN2 shows advanced and faster regrowth compared to a strain with empty vector. But a big size colony shows no changed regrowth and a small size colony shows delayed and decreased regrowth. We think the overexpression experiment is not directly relevant to the main conclusion in the manuscript, and very difficult to interpret. Given the potential artifacts and the diversity of outputs, we didn’t include these results in the manuscript.

**Author response image 1. respfig1:** Growth curves of strains overexpressing Msn2 and Msn4. Msn2 and Msn4 were cloned onto a 2 micron plasmid and transferred into a leu2Δ CEN.PK strain. We observed variability in colony sizes. Shown here are technical replicated of growth curves from a small (**S**). medium (**M**) and big (**B**) colony overexpressing Msn2 and Msn4. 6 day old stationary cultures were diluted back into YP media containing 0.1% (shown in oink) or 2% Dextrose (shown in green) and OD600 was measured over 72 hours. pRS425 is the empty vector control.

2) Msn2 and Msn4 are activated by dephosphorylation and translocation from the cytoplasm to the nucleus, therefore nuclear localization can serve as a reporter for Msn2/4 activity. In the paper, the authors only monitored the expression changes during YMC. It would serve as the direct supportive evidence for Msn2/4 activation if the authors can observe nuclear localization of Msn2/4 specifically during the RC phase. And if so, what would be the authors' speculation on the mechanisms of Msn2/4 activation? This will provide some clues on why Msn2/4 are activated specifically during the RC phase.

Besides the expression changes during YMC, we also observed the increase of ChIP-seq signals in the RC phase (Figure 2, Figure 3—figure supplement 1). Because Msn2/4 needs to move into the nucleus and bind to the genome to be active, we think the ChIP-seq results serve as direct evidence of Msn2/4 activation. We didn’t examine the mechanisms of Msn2/4 activation in this manuscript. Given that YMC is a nutrient limited condition, we speculate that TOR signaling may be involved in the activation, since TORC1 can sense nutrient availability and inhibition of TORC1 leads to nuclear translocation and activation of Msn2/4 (Beck and Hall, 1999; Loewith and Hall, 2011). We added a discussion of these points in the Discussion section.

3) The authors claim that Msn2/4 have a dual role in regulating carbohydrate metabolism genes and stress resistance genes. Under the conditions that they are focusing on (prolonged nutrient-limited condition), do the stress resistance genes also contribute to cell growth or they are unrelated by-products? An analysis or at least a discussion on this would be helpful.

We think the stress resistance genes are primarily activated to improve survivability. However, we think some of the genes may also contribute to cell growth. For example, TPS2 and GAC1 are involved in synthesis of trehalose and glycogen. They are storage carbohydrates and important for stress resistance. They have also been shown to be metabolized during the exit of quiescence to fuel growth (Shi et al., 2010; Sillje et al., 1999). We add discussions of these points in subsection “Regulatory network for Msn2/4 in YMC” and subsection “Systematic comparison between YMC and distinct quiescent states”.

Reviewer #2:This paper describes analysis of how the transcription factors Msn2 and Msn4 regulates expression of genes encoding enzymes of the central carbon metabolism in yeast, with focus on regulation during the metabolic cycle of yeast. These TFs are known to play a central role in stress regulation, and they regulate expression of a very large number of genes in yeast (>100). The authors claim that based on their findings it can be suggested that Msn2/4 control the transition from reductive charging phase to oxidative phase by regulating the accumulation of AcCoA. Regulation of shifts in yeasts metabolism is highly complex, and it tends to the naive to suggest such a statement, and it is by no means supported by the data. Thus, the authors focus their discussion on how Msn2/4 binds (and control expression) to genes of the glycolysis, but these genes are controlled by a large number of other TFs. Furthermore, Msn2/4 themselves control a large number of other genes, including several other TFs, which is not considered in their analysis. A main line of argument is measurements of AcCoA, but this metabolite is present in several cellular compartments – mitochondria, cytosol, peroxisomes and the nucleus (probably equilibrated with the cytosolic pool), and there is no free transport of AcCoA between these compartments. Measuring the total level and linking this to enzymes that are operating in different compartments is not correct (Figure 3 is wrong as ACS1 is forming acetyl-CoA in the cytosol and PDA1 is forming it in the mitochondria). The paper is interesting as the authors present a lot of data, but the biological conclusion they are trying to extract from these data is not supported by the data, and is likely wrong.

We thank the reviewer of the insightful comments. However, we respectfully disagree with the reviewer’s conclusion.

1) We agree with the reviewer that regulation of shifts in yeast metabolism is highly complex, and that is one reason that our study focuses on a single metabolite, acetyl-CoA. The concept that acetyl-CoA stimulates cell growth and proliferation has been reported in many systems such as yeast (Takahashi et al., 2006), embryonic stem cells (Moussaieff et al., 2015) and cancer cells (Donohoe et al., 2012; Lee et al., 2014; Wellen et al., 2009; Shan et al., 2014; Comerford et al., 2014) and the role of acetyl-CoA is specifically in YMC to activate growth-promoting genes (Cai et al., 2011) and cell cycle genes (Shi et al., 2013). Based on these works, our study provides a mechanistic advance about the source of acetyl-CoA in the RC of YMC. We believe this is a critical extension to help understand the exact role of acetyl-CoA in YMC and general cell growth. We did not claim that acetyl-CoA accumulation is the only route by which Msn2/4 drives the transition. We discuss the role of Msn2/4 in activating stress-response genes to maintain survival and the storage carbohydrates (trehalose and glycogen) in stress resistance and regrowth (subsections “Regulatory network for Msn2/4 in YMC” and “Systematic comparison between YMC and distinct quiescent states” and in the Discussion section.).

2) We agree with the reviewer that glycolysis genes are controlled by other TFs and Msn2/4 controls other TFs that in turn regulate glycolysis genes. We also added Figure 1 and Figure 1—figure supplement 1 to analyze the targets of these TFs. However, our ChIP-seq and RNAseq data suggests that Msn2/4 directly target these genes and regulate their expression. This doesn’t contradict the fact that other TFs may also regulate glycolysis genes. The prevalent and dramatic reduction of glycolysis transcripts in the msn2/4 mutant indicates that Msn2/4 has a significant effect on glycolysis. As the reviewer stated, other TFs that regulate glycolysis genes are also controlled by Msn2/4. We think the upstream role of Msn2/4 further support Msn2/4 as the master regulator of glycolysis.

3) We agree with the reviewer that the total level of acetyl-CoA doesn’t reflect the levels in each compartment and it may change the interpretation. We updated Figure 3 and the text and cited references to clarify the confusing aspects. (A) Although the PDH complex can generate acetyl-CoA in mitochondria, the expression level is not regulated by Msn2/4 (Figure 3, Figure 4—source data 2). On the other side, the cytosolic enzymes, Pdc1, Adh2 and Acs1 are all targeted by Msn2/4. The expression level of ALD2 is also dramatically reduced in the *msn2msn4* mutant. (B) The expression level of ACS1 is significantly induced in the RC phase but not PDA1 (PDH complex), suggesting an enhancement of the cytosolic pathway. (C) Cai et al., (2011) have shown that adding different components of the cytocolic route, like acetate, ethanol, acetaldehyde, can immediately induce the transition from RC to OX. It is unlikely that these molecules can all be transported into the mitochondria and converted to acetyl-CoA there. (D) Cai et al., (2011) have shown that during the transition, histone acetylations are strongly increased and a gcn5 (histone acetyltransferase) mutant shows a similar YMC defect, suggesting the role of the nucleo-cytosolic pool of acetyl-CoA in driving regrowth. Our transcriptional analysis is consistent with the hypothesis of a key nucleocytosolic pool of acetyl-CoA. (E) Finally, we would like to clarify that our study focuses on the mechanism of acetyl-CoA generation and not the mechanism of how acetyl-CoA stimulates growth and we agree that it will be very important to elucidate the detailed mechanisms of how acetyl-CoA stimulates growth, including the roles of each pool of acetyl-CoA, but this is beyond the scope of this paper. And we would like to emphasize that the majority of the glycolytic pathway, from glucose transportation to pyruvate, occur in the cytosol and are all controlled by Msn2/4 (Figure 3).

**Author response image 2. respfig2:** 

Reviewer #3:1) The authors could do a better job in integrating current literature on the importance of carbohydrate stores, particularly trehalose and glycogen, in survival in nutrient stress conditions and re-entry into the cell cycle from quiescence.

Thank you for the good suggestion. We have expanded the discussion and added references to emphasize their important roles in regrowth (subsection “Regulatory network for Msn2/4 in YMC”).

2) Why did the authors focus on H3K9 acetylation, rather than gene expression, in the DynaMO prediction transcription factors that drive the YMC? (This may be detailed in the accompanying manuscript. However, some more details in this manuscript would be helpful to understand the outcome of the results.) Does the focus on H3K9ac provide a better assessment of gene expression that is responsive to acetyl-CoA?

Sorry for the confusion. We added a description in the Results section to explain how the prediction works. H3K9ac not only marks the actively transcribed genes but also marks the transcription regulatory regions (promoters and enhancers). Based on the assumption that TFs bind to promoters and enhancers to function, a combination of spatial information of H3K9ac and TF motifs increases the prediction accuracy of TFBSs.

We didn’t find a special advantage of H3K9ac in predicting genes that is responsive to acetyl-CoA. We used H3K9ac because it shows the highest dynamics across YMC. Other histone acetylations show similar trends, but are a lot less dynamic and thus the signal is not as strong.

3) The Introduction states that TFs were deleted to validate their role in the YMC. This is a bit vague. Can the authors state which ones? Is Msn2 and 4 unique in terms of its YMC disruption?

We added the names of the TFs we examined in subsection “DynaMO identifies TFs associated with specific phases of YMC”. Different defects were observed in different TF mutants. Among the examined TFs, the elongated RC phase was only observed in the Msn2 and Msn4 mutants.

4) In the Introduction, the authors state that Msn2 and 4 are known to regulate stress genes. This statement should include relevant references, such as Gasch et al., 2000; Martínez-Pastor et al., 1996; Schmitt and McEntee, 1996).

We have added the corresponding references.

5) The authors may also want to discuss upstream factors that regulate Msn2 function. In particular Tor signaling, which is influence by nutrient availability.

We added a discussion about TOR signaling as a potential upstream factor that regulate Msn2/4 (Discussion section). We think it might be an interesting future direction.

[Editors’ note: the author responses to the second round of peer review follow.]

Reviewer #1:In the revision, the authors did not examine the YMC progression or acetyl-CoA level (which are where the major debates centered on) in the overexpression strain, because "the YMC requires coordination of various parameters"(?). Instead, they just checked the regrowth rate and got a variety of inconsistent results. This is not surprising given that they transformed yeast with a 2 micro plasmid (pRS425) expressing Msn2 or Msn4. The variability of plasmid copy numbers is huge, and therefore the kind of colony variability is expected no matter which output phenotypes (growth rate or others) you will be looking at. A genomic integration (with pRS305) will work way better.At this point, it seems my attempts to save the debated conclusion did not work out. However, given their striking findings summarized in my first review, I still support the publication of this manuscript, but would suggest the authors to tone down their conclusion and avoid claiming of direct causal or driving relationship.

We have revised the text including the title and text to avoid implying any direct causal relationship between glycolysis regulation and acetyl-CoA accumulation.

Reviewer #3:[…] In essence, the title of the manuscript should read something like 'Msn2 / Msn 4 regulate the expression of glycolytic enzymes during the recovery phase of the metabolic cycle'. I'm looking forward to discussing with the other reviewers whether this discovery on its own meets the bar set by eLife. I think it perhaps does; eLife did aim for very high in the past, but one needs to see this in the light of the recent and effectual situation. One might recommend to the authors however to tone down the manuscript to the part that they actually show.

We have revised the text including the title to avoid any direct causal relationship between glycolysis regulation and acetyl-CoA accumulation.

[Editors' note: revisions were requested prior to acceptance, as described below.]

In particular, please:1) Tone-down the claims concerning acetyl-coA regulation. All reviewers were of the opinion that your study does not establish a functional connection from transcription regulation of glycolysis to the mechanism leading to altered acetyl-co levels.

We have revised the text including the title to remove any explicit functional connection between the transcriptional regulation of glycolysis and altered acetyl-CoA levels. In the updated presentation, we only project that the transcription regulation of glycolysis is a potential mechanism by which acetyl-CoA is accumulated.

2) Validate the antibodies used, as requested by the new reviewer.

We have added western blot results of the WT, msn2 and msn4 lysates against Msn2 and Msn4 antibodies in Figure 2—figure supplement 1 to show the good performance of the antibodies.

3) Please see the other minor comments of the two new reviewers and perhaps emphasize some more the points that they emphasize as being mostly significant about your findings. These revisions are not essential but may improve your presentation.

We have addressed each comment of all the reviewers. Please see our point-by-point response below.

Reviewer #1[…] I would ask them to improve Figure 3—figure supplement 1, the legends/labels on the left side of the ChIP-seq views are hard to read, and also comment on the fact that there seems to be binding of these transcription factors throughout the entire coding region for some of these glycolytic genes, in addition to the promoter regions, which could be potentially interesting? or artifactual? Also, they appear to use commercial antibodies from Santa Cruz for ChIP of Msn2/4, some of their own validation should be included to help demonstrate the specificity of these antibodies (e.g., a Western of WT vs. msn2∆, msn4∆ lysates).

We have added enlarged labels on the left side of the ChIP-seq views in Figure 3—figure supplement 1. We have also described and commented on the binding signals of Msn2/4 in the coding regions of target genes. We don’t think these are artifacts because many of the genes with binding signals in the coding regions show altered expression by either RNAseq (Figure 4) or RT-qPCR (Figure 3), suggesting that Msn2/4 do regulate their expression. Coding regions of glycolytic genes have higher conservation scores in other *Saccharomyces* strains than coding regions of other genes or promoter regions of glycolytic genes (Figure 3—figure supplement 1). However, functional testing of the binding sites in the coding regions is beyond the scope of this manuscript. We have also added the western blot result in Figure 2—figure supplement 1, which demonstrate the specificity of these antibodies.

Reviewer #2[…] The importance of this paper to my mind is to broaden the appreciation of the roles of Msn2 and Msn4 from general stress resistance transcription factors to regulators of the metabolic cycle that have special roles in metabolism of cells that are coming out of quiescence that are not evident in standard growth conditions. The work is quantitative, incorporating multiple dimensions of analysis (though the magic sauce in DYNAMO was not described) and hence gives us a more comprehensive appreciation of the relevant targets of Msn2 and Msn4. As with any paper that provides a list of genes/proteins, there are some puzzles that are left unanswered such as why GIS1 is a target of this regulation. But on balance, the authors do a good job of drawing together logic behind the interactions described.

We have a short description of DynaMO in the end of introduction and beginning of the result section since it is not the main focus of this manuscript. We believe it will be a valuable tool to study transcription regulation in dynamic processes and the method paper describing DynaMO is currently under revision and will likely publish around the same time.

We described GIS1 as another TF identified by DynaMO as a candidate TF regulating RC phase and it shows similar targets as Msn2/4 (Figure 1). Whether GIS1 itself is a target of Msn2/4 is unclear. We saw binding signals of Msn2/4 at GIS1 but the expression of GIS1 was not significantly changed in the *msn2Δmsn4Δ* double mutant. Based on this negative result we didn’t include it as a core target of Msn2/4.

In searching for broader relevance, the authors speculate that Myc might be the human equivalent for the regulation of tumor growth since they assert that tumors are nutrient limited. That assertion may be true, but it certainly isn't widely known and raised my skepticism about this point, which is not that important to the results of the paper, but should be checked.

The nutrient environment of cancer cells is complex and dynamic. We have revised the words to clarify the conditions under which cancer cells survive under nutrient limited conditions. We have also added a reference that specifically demonstrates that acetyl-CoA synthetase is important to fuel cancer cells under nutrient limited conditions. We also carefully described the connection between Msn2/4 and Myc. We only speculate that Myc might be a functional equivalent of Msn2/4.

So why did I have to read it a couple times to get it? I think the authors could do a better job of emphasizing how different previous conditions used for the study of Msn2 and Msn4 are from the conditions encountered in the metabolic cycle. I would also advocate losing that damn acronyms for the three phases of the cycle as they are not commonly used and are disruptive to readers who are trying to understand this under explored region of metabolism.

We have carefully compared the previous conditions and the YMC condition in the discussion and analyzed the reason that YMC can be a good platform for revealing the roles of Msn2/4 in transitions from starvation to growth. We apologize for the difficult to remember acronyms (we totally agree!!) but these are the abbreviations used in all the other papers. To help readers understand the three phases of the cycles, we have used metaphors in the introduction to liken OX, RB and RC as growth, proliferation and quiescence. We have also sprinkled these pseudo-synonyms throughout the text to remind readers about the meanings of the three phases.